# Radon Activity Concentrations in Natural Hot Spring Water: Dose Assessment and Health Perspective

**DOI:** 10.3390/ijerph18030920

**Published:** 2021-01-21

**Authors:** Eka Djatnika Nugraha, Masahiro Hosoda, June Mellawati, Untara Untara, Ilsa Rosianna, Yuki Tamakuma, Oumar Bobbo Modibo, Chutima Kranrod, Kusdiana Kusdiana, Shinji Tokonami

**Affiliations:** 1Centre for Technology of Radiation Safety and Metrology, National Nuclear Energy Agency of Indonesia (BATAN), Jakarta 12440, Indonesia; eka.dj.n@batan.go.id (E.D.N.); june_mellawati@batan.go.id (J.M.); tara@batan.go.id (U.U.); kusdiana@batan.go.id (K.K.); 2Department of Radiation Science, Graduate School of Health Sciences, Hirosaki University, Hirosaki 036-8504, Japan; m_hosoda@hirosaki-u.ac.jp (M.H.); tamakuma@hirosaki-u.ac.jp (Y.T.); h19gg701@hirosaki-u.ac.jp (O.B.M.); 3Institute of Radiation Emergency Medicine, Hirosaki University, Hirosaki 036-8504, Japan; kranrodc@hirosaki-u.ac.jp; 4Centre for Nuclear Minerals Technology, National Nuclear Energy Agency of Indonesia (BATAN), Jakarta 12440, Indonesia; ilsa.r@batan.go.id

**Keywords:** radon, hot spring, dose assessment, public health

## Abstract

The world community has long used natural hot springs for tourist and medicinal purposes. In Indonesia, the province of West Java, which is naturally surrounded by volcanoes, is the main destination for hot spring tourism. This paper is the first report on radon measurements in tourism natural hot spring water in Indonesia as part of radiation protection for public health. The purpose of this paper is to study the contribution of radon doses from natural hot spring water and thereby facilitate radiation protection for public health. A total of 18 water samples were measured with an electrostatic collection type radon monitor (RAD7, Durridge Co., USA). The concentration of radon in natural hot spring water samples in the West Java region, Indonesia ranges from 0.26 to 31 Bq L^−1^. An estimate of the annual effective dose in the natural hot spring water area ranges from 0.51 to 0.71 mSv with a mean of 0.60 mSv for workers. Meanwhile, the annual effective dose for the public ranges from 0.10 to 0.14 mSv with an average of 0.12 mSv. This value is within the range of the average committed effective dose from inhalation and terrestrial radiation for the general public, 1.7 mSv annually.

## 1. Introduction

Odourless and originating from radium-226 (^226^Ra) decay that naturally occurs in the earth’s crust, radon is a radioactive noble gas. According to the United Nations Scientific Committee on the Effects of Atomic Radiation (UNSCEAR), half of the world’s mean value of annual effective dose by natural radiation sources is attributed to ^222^Rn, thoron (^220^Rn) and their progenies [1]. Radon (^222^Rn) has been recognised as a carcinogenic gas and is well-known as the second leading health risk factor for lung cancer [1,2,3]. Radon from water contributes to the total inhalation risk associated with radon in indoor air. In addition to this, drinking water contains dissolved radon and the radiation emitted by radon and its radioactive decay products exposes sensitive cells in the stomach as well as other organs once it is absorbed into the bloodstream. Noting this danger, the United States Environmental Protection Agency (EPA) proposed a maximum contaminant level (MCL) for radon in the water around 11 Bq L^−1^ [4]. 

Radon dissolves in water that passes through soil and rock containing the natural radioactive substance [5,6]. As a result, water moving deeper through the earth’s crust gathers increasing concentrations of radon and other natural radioactive materials. When, during the geothermal process, temperatures and pressures increase enough, some of this water is expelled through faults and cracks, reaching the earth’s surface as hot springs. Hot spring water produced under these circumstances usually contains high concentrations of ^222^Rn. This is due to at least one of two natural processes: ^226^Ra dissolving in the water after interacting with rock and soil in the earth or ^222^Rn entering the water from rocks containing ^226^Ra [6,7,8].

The world community has long used natural hot springs for tourist and medicinal purposes. In Indonesia, the province of West Java, which is naturally surrounded by volcanoes, is a prime hot spring tourist destination. Approximately 1.8 million tourists visit natural hot springs in the West Java province each year [9,10,11].

It is, therefore, necessary to study the contribution of ^222^Rn doses from natural hot spring water as part of radiation protection for public health. This paper is the first report on ^222^Rn measurements in tourism natural hot spring water in Indonesia. Previous studies related to radon measurements in Indonesia included measurements of air at dwellings [12,13,14,15], ^222^Rn in water samples [15], ^222^Rn in geothermal and geosciences [16]

## 2. Materials and Methods 

### 2.1. Study Area

This research was conducted in several districts in West Java, including the Ciater Hot Springs area in Subang; the Ciwidey and Pangalengan Hot Springs areas in Bandung; and the Cipanas and Darajat Hot Springs areas in Garut. Each of these hot springs is a major tourist destination, as shown in Figure 1. Visited by approximately 300,000 tourists each year, the Ciwidey and Pangalengan areas are tourist destinations located on the Patuha volcano. The Ciater Hot Spring, located on the Tangkuban Perahu volcano, is the most popular area for hot spring tourism with approximately 1.3 million visitors annually. Finally, as many as 50,000 tourists visit the Cipanas and Darajat areas every year [9,10,11]. The Cipanas area is located on the Guntur volcano, and the Darajat area is on the Kamojang volcano, which also has a geothermal power plant. 

### 2.2. Radon Measurement in Water Samples

A total of 18 water samples of 250 mL each were collected using radon-tight reagent bottles as part of the water analysis accessory (RAD-H2O, Durridge Co., USA). This study was conducted in September 2019, which includes the dry season. The samples included 17 natural hot springs water samples and one mineral water sample. The samples were measured for temperature, pH, and electroconductivity (E.C.) (Laquatwin, Horiba, Japan). In addition, the ambient dose equivalent rates (PDR-111, Hitachi, Japan) around the sampling area were measured. An electrostatic collection type ^222^Rn monitor (RAD7, Durridge Co., USA) connected to a water analysis accessory was used to measure the samples and detect alpha activity. The RAD7 detector connected the monitor with a bubbling kit for degassing of ^222^Rn in a water sample into the air in a closed circuit, as shown in Figure 2. Before the ^222^Rn arrived at the detector, it also needed to be dried with a desiccant (CaSO_4_, Drierite, W A Hammond, USA) to absorb the moisture. 

We used the WAT250 protocol in five-minute cycles and five recycles to generate data. In this measurement protocol for grab samples analysis, the pump ran for five minutes, flushing the measurement chamber, and then stopping. The RAD7 waited for five additional minutes at the end of the run before printing a summary. Since the analysis was made more than an hour after the sample was taken, a correction was applied to account for ^222^Rn decay [17]. The amount of radon loss was calculated using the decay formula, or Equation (1):(1)$$C_{t}^{\prime }=\;C_{0}^{\prime }Ae^{-\lambda t^{\prime }/60}$$
where $C_{t}^{\prime }$ (Bq m^−3^) is the ^222^Rn activity concentration at time *t*′ (min); $C_{0}^{\prime }\;$ is the ^222^Rn activity concentration at time *t*′ = 0; and *λ* is the ^222^Rn decay constant (7.542 × 10^−3^ h^−1^). 

### 2.3. Radon Measurement in the Air

We measured the ^222^Rn activity concentration in the air 1 m above the hot spring pool with RAD7 for 8 h. An ‘auto’ mode was used to obtain this measurement in 60 min cycles, and eight recycles were allowed. ^222^Rn measurements began with the sniff mode before changing automatically to the normal mode after 3 h 45 min. The results obtained were then averaged. We also measured the ^222^Rn activity concentration in the dwelling around each natural hot spring area as the background for estimating the transfer coefficient from ^222^Rn in the water to ^222^Rn in the air.

### 2.4. Estimation of Annual Effective Dose

We used Equation (2) to calculate the contribution of ^222^Rn in the water to ^222^Rn in the atmosphere. Meanwhile, the internal annual effective dose from ^222^Rn through inhalation, the annual effective dose from external radiation, and the annual effective dose are shown in Equations (3)–(5), respectively.
(2)$$D_{Rn-w}=C_{Rn-w}\times TF_{Rn-w-a},$$
(3)$$E_{in\left(Rn\right)}=C_{Rn}\times F\times DCF_{RnP}\times T,$$
(4)$$E_{ext}=H^{*}\times \mathrm{CF}\times DCF_{H-D}\times T,$$
(5)$$\mathrm{AED}=E_{ext}+E_{in-Rn}.$$

In Equation (2), $D_{Rn-w}\;$ is the ^222^Rn activity concentration contributed from water to the atmosphere (Bq m^−3^); $C_{Rn-w}$ is the ^222^Rn activity concentration in the water samples (Bq L^−1^); and $TF_{Rn-w-a}$ is the transfer coefficient from water to air, which equals 1 × 10^−4^ [18,19]. In Equation (3), $E_{in\left(Rn\right)}\;$ is the internal annual effective dose from ^222^Rn through inhalation (mSv); $C_{Rn}\;$ is the ^222^Rn activity concentration in the air (Bq m^−3^); F is the equilibrium factor of ^222^Rn and radon progeny, which equals 0.4; $DCF_{RnP}$ is the dose conversion factor for ^222^Rn, which equals 1.7 × 10^−5^ mSv (Bq h m^−3^)^−1^ [3,20]; and T is the time, which is 2000 h for the worker and 8 h a week, or 384 h annually, for the public. In Equation (4), $E_{ext}$ is the annual effective dose from external radiation; $H^{*}$ is the ambient dose equivalent rates (nSv h^−1^); CF is the conversion factor from ambient dose equivalent rates to the absorbed dose in the air, which equals 0.652 (Gy Sv^−1^) [21]; and $DCF_{H-D}$ is the conversion factor from the absorbed dose in the air to the external effective dose, which equals 0.7 [1]. Finally, AED is the annual effective dose (mSv).

## 3. Results and Discussion

Natural hot springs are a popular tourist attraction in West Java. Since at least 1980, tourists have enjoyed the natural atmosphere of the area while participating in activities such as swimming, soaking, photographing the scenery, and walking in the park. Many even stay overnight [9,10,11]. Despite its diverse other uses, natural hot spring water in West Java is not used for drinking. Therefore, we performed dose assessments through inhalation and external dose radiation only.

From a total of 17 natural hot spring water samples and one mineral bottled water sample, the value of electroconductivity ranged from 0.164 to 1.925 mS cm^−1^ with an average value of 1.541 mS cm^−1^. Meanwhile, the pH ranged from 5 to 7 with an average value of 6. Finally, water temperature, as shown in Table 1, ranged from 36 to 42 °C with an average of 39 °C. According to the regulations of the Indonesian Ministry of Health [22], and the World Health Organisation [23], the E.C. and pH values in natural hot spring water samples in West Java fall above recommended values, which must be below 1.5 mS cm^−1^ and 6.5–8.5 for E.C. and pH, respectively. E.C. is closely related to the content of dissolved solids in the water. Thus, if water with a high E.C. value and pH level is used for drinking, gastrointestinal upset and kidney disease can result. Unlike the natural hot spring water samples, the mineral bottled water is suitable for drinking.

The dissolved ^222^Rn in water samples in the natural hot spring area shown in Figure 3 have a range of 1 to 31 Bq L^−1^. With the exceptions of water samples A1, B4, and C3, these values were all below the maximum concentration limit (MCL), 11 Bq L^−1^, suggested by the EPA. Water samples from this natural hot spring area contained dissolved ^222^Rn activity concentrations higher than the MCL but within the limit of the alternative maximum concentration level (AMCL) of 148 Bq L^−1^, also suggested by the EPA [4]. Based on the UNSCEAR 2000 report, the AMCL of 148 Bq L^−1^ is the limit determining the concentration of ^222^Rn in the water that will produce an indoor ^222^Rn increment equal to an outdoor ^222^Rn activity concentration of 15 Bq m^−3^ with the transfer coefficient from water to indoor air applied as 1 × 10^−4^ [18]. The ^222^Rn activity concentration in natural hot spring water in the West Java province will contribute to an ^222^Rn activity concentration in the air equal to 0.1–3.1 Bq m^−3^ concurrent to the ^222^Rn activity concentration in air measured in this study as shown in Table 2.

Comparing the values of ^222^Rn levels in West Java to ^222^Rn activity concentrations elsewhere reveal that Indonesian levels are rather low. Studies in Slovenia [24], the U.S. [25], Spain [26], Taiwan [27], Hungary [28], Poland [29], Venezuela [30], Germany [31], Croatia [32], Iran [33], and Thailand [6] report ^222^Rn activity concentrations in hot spring water ranging from 0.2 to 600 Bq L^−1^. The radon concentration in an area is closely related to geological rock types, which in West Java have andesitic rock types that contain low uranium and radium content [34,35,36].

The radon activity concentration in the air in the hot spring area of the West Java province ranges from 35 to 50 Bq m^−3^ with an average of 42 Bq m^−3^. Equation (6) compares the activity value of the ^222^Rn activity concentration dissolved in water and the ^222^Rn in the air.
(6)$$\mathrm{Transfer}\;\mathrm{coefficient}=\frac{\bar{\Delta C_{a}}}{\bar{C_{w}}},$$
here, $\Delta C_{a}$ is the average increment of ^222^Rn activity concentration in the air (Bq m^−3^). This result is a subtraction of the ^222^Rn activity concentration in the air around the natural hot spring pool from the ^222^Rn activity concentration outside of dwellings around the hot spring area. The ^222^Rn activity concentration outside of dwellings around the hot spring area for Cipanas, Darajat, Ciwidey, Pangalengan and Ciater were 30, 30, 28, 32, and 35 Bq m^−3^, respectively. $C_{w}$ is the dissolved ^222^Rn activity concentration in the water (Bq m^−3^). The value of ^222^Rn coefficient transfer from water to air in this study was an average of 2.0 × 10^−03^. This value is higher than the value UNSCEAR [18] and Hopke et al. [19] found, possibly due to the effect of water mixing, since tourists were active in the pool while we conducted measurements. Others, including Radolic et al. [32] and Song et al. [37], have reported average transfer coefficients around 4.9 × 10^−3^ and 1.5 × 10^−3^, respectively.

The annual effective dose in the natural hot spring water area ranges from 0.51 to 0.71 mSv with a mean of 0.60 mSv for workers. Meanwhile, the public dose ranges from 0.10 to 0.14 mSv with an average of 0.12 mSv. This value falls within the average committed effective dose from inhalation and terrestrial radiation for the general public, 1.7 mSv annually, determined by UNSCEAR [1].

## 4. Conclusions

The concentration of ^222^Rn in natural hot spring water samples in the West Java region of Indonesia has a range of 1 to 31 Bq L^−1^. An estimate of the annual effective dose in the natural hot spring water area ranges from 0.51 to 0.71 mSv with a mean of 0.60 mSv for workers. Meanwhile, the public is exposed to a range of 0.10 to 0.14 mSv with an average of 0.12 mSv. This value falls within the range of the averaged committed effective dose from inhalation and terrestrial radiation for the general public, 1.7 mSv annually.

## Figures and Tables

**Figure 1 ijerph-18-00920-f001:**
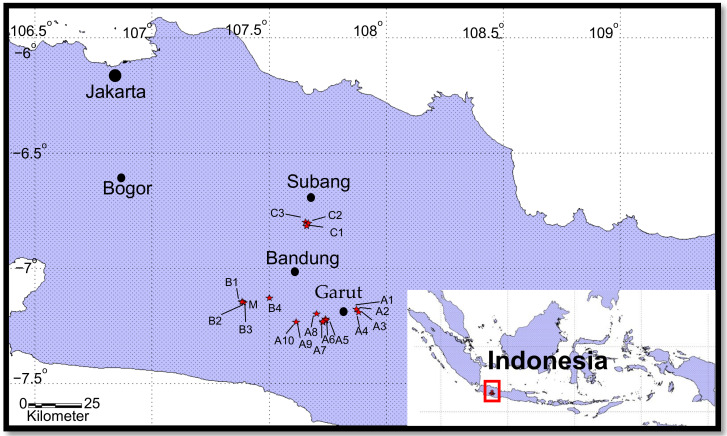
The study area, covering three cities: Bandung, Subang, and Garut. The black dots represent cities, and red asterisks indicate sampling locations.

**Figure 2 ijerph-18-00920-f002:**
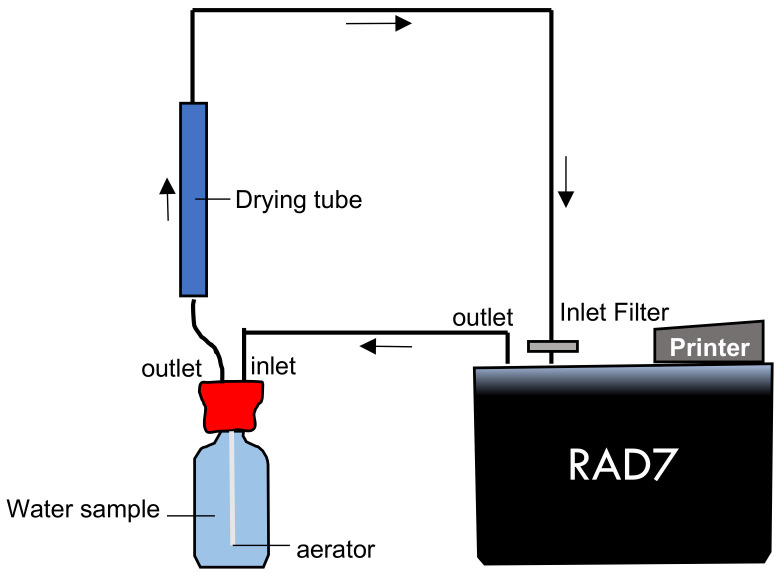
Schematic of experimental setup for measuring dissolved ^222^Rn measurement

**Figure 3 ijerph-18-00920-f003:**
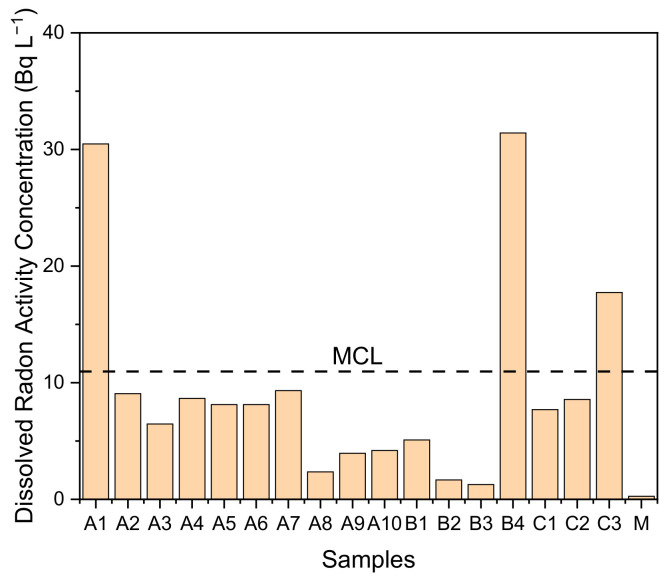
Radon activity concentration in water samples. MCL = maximum contaminant level.

**Table 1 ijerph-18-00920-t001:** The location, physical and chemical properties of the water samples.

Samples	Area	Longitude	Latitude	Elevation	Temperature	Electroconductivity at 25 °C	pH
		(E)	(S)	(m)	(°C)	(mS cm^−1^)	
A1	Cipanas	107.8716	−7.17643	1678	38	1.511	6
A2	Cipanas	107.8816	−7.18884	1675	38	1.459	6
A3	Cipanas	107.7016	−7.19645	1668	39	1.485	6
A4	Cipanas	107.5016	−7.19646	1671	39	1.458	6
A5	Darajat	107.7415	−7.21833	1672	37	1.442	6
A6	Darajat	107.7414	−7.21914	1670	38	1.538	6
A7	Darajat	107.7416	−7.22191	1672	37	1.586	6
A8	Darajat	107.7287	−7.22935	1973	42	1.682	5
A9	Darajat	107.7287	−7.22906	1976	42	1.628	5
A10	Darajat	107.7287	−7.22851	1985	42	1.677	5
B1	Ciwidey	107.3843	−7.14416	1779	37	1.425	6
B2	Ciwidey	107.3901	−7.14710	1781	39	1.590	6
B3	Ciwidey	107.3853	−7.14429	1724	36	1.385	6
B4	Pangalengan	107.6148	−7.23211	1450	39	1.925	5
C1	Ciater	107.6544	−6.80861	873	36	1.401	6
C2	Ciater	107.6544	−6.80861	885	38	1.415	6
C3	Ciater	107.6544	−6.80862	897	39	1.598	5
M	Bottled water	-	-	-	25	0.164	7
min					36	0.164	5
max					42	1.925	7
average					39	1.541	6

**Table 2 ijerph-18-00920-t002:** Details of measuring result: Dissolved ^222^Rn in water, ^222^Rn in air, and calculation of annual effective dose.

Samples	Dissolved Radon in Water	Radon in Air	Ambient Dose Equivalent Rate	Radon Transfer Coefficient from Water to Air	Contributed Dissolved Radon in Water to Radon in Air	Annual Effective Dose due to Ingestion (mSv)		Annual Effective Dose due to External Exposure (mSv)		Total Annual Effective Dose (mSv)	
	Bq L^−1^	Bq m^−3^	nSv h^−1^		Bq m^−3^	Worker	Public	Worker	Public	Bq L^−1^	Bq m^−3^
A1	31 ± 3.4	48 ± 7	43 ± 2	5.9 × 10^−4^	3.1	0.65	0.13	0.04	0.01	0.69	0.13
A2	9 ± 1.0	49 ± 7	41 ± 2	2.1 × 10^−3^	0.9	0.67	0.13	0.04	0.01	0.70	0.14
A3	7 ± 0.7	42 ± 6	43 ± 2	1.8 × 10^−3^	0.7	0.57	0.11	0.04	0.01	0.61	0.12
A4	9 ± 1.0	38 ± 6	41 ± 2	9.3 × 10^−4^	0.9	0.52	0.10	0.04	0.01	0.55	0.11
A5	8 ± 0.9	39 ± 6	38 ± 2	1.1 × 10^−3^	0.8	0.53	0.10	0.03	0.01	0.57	0.11
A6	8 ± 0.9	39 ± 6	37 ± 2	1.1 × 10^−3^	0.8	0.53	0.10	0.03	0.01	0.56	0.11
A7	9 ± 1.0	40 ± 6	36 ± 2	1.1 × 10^−3^	0.9	0.54	0.10	0.03	0.01	0.58	0.11
A8	2 ± 0.3	38 ± 6	40 ± 2	3.4 × 10^−3^	0.3	0.52	0.10	0.04	0.01	0.55	0.11
A9	4 ± 0.4	38 ± 6	41 ± 2	2.0 × 10^−3^	0.4	0.52	0.10	0.04	0.01	0.55	0.11
A10	4 ± 0.5	38 ± 6	40 ± 2	1.9 × 10^−3^	0.4	0.52	0.10	0.04	0.01	0.55	0.11
B1	5 ± 0.6	38 ± 6	36 ± 2	2.0 × 10^−3^	0.5	0.52	0.10	0.03	0.01	0.55	0.11
B2	2 ± 0.2	37 ± 6	38 ± 2	5.4 × 10^−3^	0.2	0.50	0.10	0.03	0.01	0.54	0.10
B3	1 ± 0.1	35 ± 6	35 ± 2	5.5 × 10^−3^	0.1	0.48	0.09	0.03	0.01	0.51	0.10
B4	31 ± 3.5	42 ± 7	44 ± 2	3.2 × 10^−4^	3.1	0.57	0.11	0.04	0.01	0.61	0.12
C1	8 ± 0.9	48 ± 7	38 ± 2	1.7 × 10^−3^	0.8	0.65	0.13	0.03	0.01	0.69	0.13
C2	8 ± 0.9	49 ± 8	37 ± 2	1.6 × 10^−3^	0.9	0.67	0.13	0.03	0.01	0.70	0.13
C3	18 ± 2.0	50 ± 8	38 ± 2	2.0 × 10^−3^	1.8	0.68	0.13	0.03	0.01	0.71	0.14
M	0.3 ± 0.1	-	-	-	0.1	-					
Average	9 ± 1.0	42 ± 6	39 ± 2	2.0 × 10^−3^	0.9	0.53	0.11	0.04	0.01	0.60	0.12
min	1 ± 0.1	35 ± 5	35 ± 2	3.2 × 10^−4^	0.1	0.48	0.09	0.03	0.01	0.51	0.10
max	31 ± 3.5	50 ± 8	44 ± 2	5.5 × 10^−3^	3.1	0.68	0.13	0.04	0.01	0.71	0.14

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
