# Peer review of "Radon Activity Concentrations in Natural Hot Spring Water: Dose Assessment and Health Perspective"

_ijerph, 2021, doi:10.3390/ijerph18030920_

Round 1

Reviewer 1 Report

Congratulations, I really enjoyed reading this article.

In my opinion, it is almost ready to be published.

I just recommend improving the introductory section adding some information about the importance of the subject and the risks involved in human health of having radon in water.

I would like to see as well a comparative analogy to other studies conducted in other regions of the world and the consequences or mitigation measures adopted.

Author Response

Ans. Thank you very much for reading our manuscript and for giving us the comments for improvements. I added the sentences as you suggest in line 38-42 "Radon from water contributes to the total inhalation risk associated with radon in indoor air. In addition to this, drinking water contains dissolved radon and the radiation emitted by radon and its radioactive decay products exposes sensitive cells in the stomach as well as other organs once it is absorbed into the bloodstream." And for comparative with another study, it already mentions in line 167-171. Thank you.

Reviewer 2 Report

This paper is interesting as for the results of measurements and as for the dose calculations presented.

It is well written and the data are presented in a good form.

Only one question about the radon-tight reagent bottles: can you explain or describe in some detail?

Author Response

Ans. Thank you very much for reading our manuscript and for giving us the comments for improvements. We use the 250mL radon tight reagent bottle that is part of the water analysis accessory (RAD-H2O, Durridge Co., U.S.A.). Based on your suggestion, we add the description to the paper as "A total of 18 water samples of 250 mL each were collected using radon-tight reagent bottles as part of the water analysis accessory (RAD-H2O, Durridge Co., U.S.A.)."

Reviewer 3 Report

Dear Authors

please check my revisions and comments in the manuscript. 

Author Response

Thank you very much for your constructive comments on our manuscript. We have read your comments carefully and response one by one

Line 19-20: "I think this is not correct, please check in MDPI Geosciences Journal, Tampomas Hot springs water and also soil gas Rn were investigated and reported in the journal "Characteristic and Mixing Mechanisms of Thermal Fluid at the Tampomas Volcano, West Java, Using Hydrogeochemistry, Stable Isotope and 222Rn Analyses " you may read and cite it. "

Ans. Thank you very much for your constructive comments on our manuscript. The topic of our paper is radon in natural hot springs at tourism areas in relation to public health for public radiation protection. We understand that in Indonesia there are many geothermal areas, but our focus is only on the regions. The sampling area is not in the source of hot springs, but in recreational pools. To avoid misinterpretation, we changed the sentence to be " This paper is the first report on radon measurements in tourism natural hot spring water in Indonesia as part of radiation protection for public health ". In addition, I cite the paper which you suggested as the radon measurement in another field (geoscience). Thank you.

Line 53-54: "Please check in previous comments"

Ans. Thank you very much for your constructive comments on our manuscript. The answer was as same as the previous question. Thank you.

Line 75: When the sampling or measurement were conducted? in dry or rainy season? As far as I know the rainfall intensity in West Java area is high especially in mountainous area. I consider some dilution of rainwater.

Ans. Thank you very much for your constructive comments on our manuscript. Yes, we agree with your opinion regarding the dilution effect. The measurement was conducted in the dry season, in September 2019. We added some information in line 81-82 " This study was conducted in September 2019, which includes in the dry season."

Line 76-77: Why you choose mineral water, in my opinion it could be better if you measured natural water e.g. cold springs or rain water as comparison, mineral water could be change chemical and some gasses may release during bottling or processing. Better compare natural hot spring vs natural water.

Ans. Thank you very much for your constructive comments on our manuscript. We agree with your opinion, that it is better to use springs water for comparison for the background.  Unfortunately, in this time we use the mineral bottled water as the background in the nearby tourism area because the residents do not always have well water. Also many residents and tourists drink the mineral bottled water as well. For this study, we think the mineral bottled water is quite representative of the public dose assessment samples. Moreover, for mechanism in geosciences,  it will be not enough. In addition, near the Ciater area is have a mineral bottled water factory, We will be considered in our future works. Thank you.

Line 101: I suggest it will be changed to 3 hour 45 minutes

Ans. Thank you very much for your constructive comments on our manuscript. I have changed it to 3 hours 45 minutes

Line 143, Table 1: I am sure this area is the only area represent sample at the northern part of your research area. In geothermal system Ciater is near the upflow zone and usually pH is arround 3 to 4, is the pH = 5 correct?

Ans. Thank you very much for your constructive comments on our manuscript. We agree with your opinion, but our sampling area is not the source in hot springs but that in recreational pools. So, it is different. Thank you.

Table 2: In previous text you mention the range is between 0.26, I guess 0.3 in this table is referred to 0.26? and it is from mineral water. So, the dissolved 222-Rn, in natural water hot springs is more than 1 Bq/L? When you make average value, you should not put mineral water value in your calculation.

Ans. Thank you very much for your constructive comments on our manuscript. We agree with your opinion and we already just including the hot spring water. Then, the min is 1 Bq/L. Thank you.

Line 182-I86: suggest you cited some papers related geogenic and geothermal radon, especially in Indonesia. Why Rn in dissolved Rn in hot springs in West Java rather low (line 162-163). I think it has strong relation with 1) source of heat or rock in Indonesia is not rich with Ra and or U, since the basement or source is not granitic rock, but here in your area is an andesitic or basaltic rock with low Ra or content. Or 2) Some radon are release or degassing since caused by temp and low permeability of overburden material (clay) clay in tropical region is thick usually. You can put some discussion part.

Ans. Thank you very much for your constructive comments on our manuscript. I added some reference regarding this matter. The sentences will be " The concentration of radon in an area is closely related to geological rock types, which in West Java have andesitic and basaltic rock types that contain low uranium and radium content."

  1. Sunarwan, B. Physical characterization of groundwater and identification of springs in the volcanic sediment aquifer (case study: Tangkuban perahu volcanic sediment in the Bandung basin) (karakterisasi phisik air tanah dan identifikasi pemunculan mata air pada akuifer endapan gunung api (studi kasus: endapan gunungapi Tangkuban perahu di cekungan Bandung). Tech journal- sci mag UNPAK, 2014, 15(1), 16-26
  2. Hidayat, M. R., Mardiana, U., Suganda, B. R., & Hadian, M. S. D. (2017). Geometry activities of Bandung area and surroundings, West Java Province (geometri akifer daerah Bandung dan sekitarnya, Provinsi Jawa Barat). Padjajaran univ Geo sci J, 2017, 1(1), 86-97.
  3. Utami, Pri. Characteristics of the Kamojang geothermal reservoir (West Java) as revealed by its hydrothermal alteration mineralogy. In Proceedings world geothermal congress 2000. Sendai, Japan 28 May – 10 June 2000, 1921-1926

Line 188-189:I think 0.26 is not representative of natural hot spring, in table 2, it was 0.3 code M and it is mineral water right?

Ans. Thank you very much for your constructive comments on our manuscript. We agree with your opinion and we already just including the hot spring water. Then the range is 1-31 Bq/L

Line 191: I think this paper is good for public health concern, but I suggest you to measure (may be next time) cold springs around volcanoes, in my experience measuring dissolved Rn in water, some times or some cold springs may have higher concentration than the hot springs water. As we know solubility of Rn in water will decreasing in higher temp. But some cold springs also bring Rn from volcanic gas, and dissolved as dissolved gas Rn.

Ans. Thank you very much for reading our manuscript and for giving us the comments for improvements. Your suggestion will be considered in our future work.

Reference 10: I am not sure

Ans. Thank you very much for your constructive comments on our manuscript. After I check it again, there have two versions, the thesis in 2013 and journal in 2013. I have changed the citation to the journal. Thank you.

  • Ananda, R.P; Ahman, E; Riwanudin, O. The effect of physical evidence of Ciwalini hot springs on the decision to visit tourists (Pengaruh physical evidence pemandian air panas ciwalini terhadap keputusan berkunjung wisatawan). Tourism and hosp essent J. 2013, vol III (1), 461.

Reviewer 4 Report

This is an old subject which became a subject of interest nowadays in the context of Radon map. It could be interesting to have some long term study in the area of pathological aspects induced in workers, considering that as professional exposure. Are they monitorized?  

Author Response

Thank you very much for reading our manuscript and for giving us the comments for improvements. The Indonesian government was conducted indoor radon measurement for the long term since 2014-2021 as the national survey. We hope soon it will be published some paper on this study. Thank you.

Round 2

Reviewer 3 Report

Dear Authors, 

I suggest revising the title by adding "case study in West Java, Indonesia" because of course not all hot springs have same situation with results in this paper. 

Best Regards, 

Author Response

Dear Authors, 

I suggest revising the title by adding "case study in West Java, Indonesia" because of course not all hot springs have same situation with results in this paper. 

 Ans: Thank you very much for your constructive comments and suggestions. Regarding the title, we would like to emphasize the proposed methodology and dose assessment of radon activity concentrations in natural hot springs, particularly tourist areas. Therefore, we will take an option on the general title. The study area we chose is West Java because West Java has a huge number of tourists for hot spring tourism. In addition, specific districts have been clearly described in the study area of the paper. We will therefore continue to use the title "Radon Activity Concentrations in Natural Hot Springs Water: Dose Assessment and Health Perspectives." thank you. We hope you will understand our intention